# A New Constructive Criterion for Markov Equivalence of MAGs

**Marcel Wienöbst**[1]            **Max Bannach**[1]            **Maciej Liśkiewicz**[1]

[1]Institute for Theoretical Computer Science , University of Lübeck , Germany

## Abstract

Ancestral graphs are an important tool for encoding causal knowledge as they represent uncertainty about the presence of latent confounding and selection bias, and they can be inferred from data. As for other graphical models, several maximal ancestral graphs (MAGs) may encode the same statistical information in the form of conditional independencies. Such MAGs are said to be *Markov equivalent*. This work concerns graphical characterizations and computational aspects of Markov equivalence between MAGs. These issues have been studied in past years leading to several criteria and methods to test Markov equivalence. The state-of-the-art algorithm, provided by Hu and Evans [UAI 2020], runs in time $O(n^5)$ for instances with $n$ vertices. We propose a new constructive graphical criterion for the Markov equivalence of MAGs, which allows us to develop a practically effective equivalence test with worst-case runtime $O(n^3)$. Additionally, our criterion is expressed in terms of natural graphical concepts, which is of independent value.

## 1   INTRODUCTION

Graphical causal models represent random variables as vertices of a graph and express causal effects of one variable on another with edges. Using the graphical approach allows an intuitive formalism to explore complex causal phenomena Spirtes et al. [2000], Pearl [2009], Koller and Friedman [2009]. Another strength of this approach is the ability to tackle causal problems using algorithmic tools, paving the way towards automated causal inference and data science.

A popular and commonly used model to encode causal knowledge, which can be inferred from data, is a *directed acyclic graph* (DAG). A DAG can be learned from conditional independence (CI) statements, if one assumes faith-

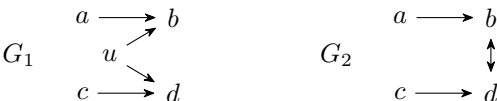

Figure 1: DAG $G_1$ encodes the CIs $X_a \perp\!\!\!\perp \{X_c, X_d\}$ and $X_c \perp\!\!\!\perp \{X_a, X_b\}$ among observed variables ($u$ is a latent variable). $G_2$ is a MAG which encodes the same CIs. The example is from [Richardson and Spirtes, 2002, Fig. 10].

fulness, that is, if the CIs among the variables are equal to those expressed through $d$-separations in the DAG Spirtes et al. [2000]. However, multiple DAGs can imply the same CI statements. For instance, if for the variables $X_a, X_b, X_c$ the only CI relationship is $X_a \perp\!\!\!\perp X_c \mid X_b$, then there are three DAGs $a \to b \to c$, $a \leftarrow b \to c$, and $a \leftarrow b \leftarrow c$, which encode the CI. We say that such DAGs are *Markov equivalent* and that they belong to the same *Markov equivalence class* (MEC). Markov equivalent DAGs encode the same conditional independances via $d$-separations and are, thus, indistinguishable on the basis of observational CIs alone.

Key results for these concepts are the graphical criterion for two DAGs to be Markov equivalent [Verma and Pearl, 1990, Frydenberg, 1990] and the graph-theoretic characterization of MECs as so-called CPDAGs [Andersson et al., 1997]. Subsequent work in this field resulted in further achievements, e. g., regarding causal structure identification from data [Meek, 1997, Spirtes et al., 2000, Chickering, 2002a,b] or causal inference and analysis based on Markov equivalence classes [Maathuis et al., 2009, van der Zander and Liśkiewicz, 2016, Wienöbst et al., 2021b]. However, things are more complicated when hidden and selective variables emerge – as is often the case in practice. Useful in this setting are (maximal) ancestral graphs (AGs, MAGs) introduced by Richardson and Spirtes [2002], which can represent uncertainty about the presence of latent confounding and selection bias, and which can be inferred from data. A variable is latent if it is not measured or recorded. For ex-

*Accepted for the 38th Conference on Uncertainty in Artificial Intelligence* (UAI 2022).

ample, the DAG $G_1$ in Fig. 1 shows a causal structure over four observed variables represented as vertices $a, b, c, d$ and a latent variable represented as $u$. $G_1$ implies the independence relations $X_a \perp\!\!\!\perp \{X_c, X_d\}$ and $X_c \perp\!\!\!\perp \{X_a, X_b\}$ over the observed variables, i.e., after marginalizing variable $X_u$ out. However, there is no DAG representing precisely these CIs, which shows that DAGs are not closed under marginalization. One can represent the CIs using MAGs as shown by $G_2$ in Fig. 1. Additionally, DAGs are not expressive enough for selection variables, which are unmeasured variables determining whether a measured unit is included in the data. Hence, DAGs are not closed under conditioning. In contrast, the class of independence models associated with AGs, i.e., the smallest class that contains the DAG independence models, is closed under marginalizing and conditioning (see [Richardson and Spirtes, 2002] for details).

Despite many advances, a number of fundamental problems concerning the properties and algorithmic aspects of this important model class remain to be explored. We investigate the Markov equivalence of MAGs – one of the basic problems in this field. As for DAGs, MAGs that encode the same conditional independencies are said to be Markov equivalent. In graphical language, we express CIs via $m$-*separations*, an extended form of $d$-separation in DAGs (formal definitions are provided in Section 2).

An effective polynomial-time algorithm to test whether two MAGs are Markov equivalent has been the subject of intense research. A naïve implementation of the definition requires testing $m$-separation relations over all pairs of vertices and all subsets of vertices, which takes exponential time. The first graphical criterion was given by Spirtes and Richardson [1996]: The *Spirtes and Richardson Criterion (SRC)* extends the conditions by Verma and Pearl [1990] and Frydenberg [1990] for DAGs and is based on the useful concept of *discriminating paths*. The SRC is intuitive and forms the basis of subsequent work. However, testing the SRC naively requires exponential time since there can be exponentially many discriminating path, which all have to be inspected. Zhao et al. [2005] proposed another characterization using the concept of minimal collider paths, which also did not lead to polynomial time. The first criterion that can be checked in polynomial time has been proposed by Ali et al. [2009]. The complexity of their method is bounded by $O(n \cdot m^4)$ for MAGs with $n$ vertices and $m$ edges. Recently, a criterion based on parametrizing sets was proposed by Hu and Evans [2020]. These sets can be generated in time $O(n \cdot m^2)$ (for dense graphs with $m \in \Omega(n^2)$ this equates to $O(n^5)$) leading to a faster algorithm.

The main contribution of this paper is a new criterion for the Markov equivalence of MAGs. It is a simple and constructive variant of the SRC and allows us to develop an algorithm for equivalence testing in cubic time. This breaks the previous $O(n^5)$ worst-case time barrier. Our criterion, coined *constructive-SRC*, is based on discriminating paths, but it

avoids searching through exponentially many paths and boils down to a simple graphical condition. The constructive-SRC is intuitive and checking it by hand is convenient. For sparse graphs with maximal degree $\Delta$, which are common in causal modeling, the running time is bounded by $O(n \cdot \Delta^2)$. We compare our algorithm experimentally with the algorithm by Hu and Evans [2020] and show that the theoretical improvements lead to better practical performance.

Obtaining the cubic runtime raises the question of whether further improvements are possible, e.g., whether a runtime of $O(n^2)$ can be attained. We discuss this issue by relating it to the Markov equivalence of DAGs, where such a runtime is achievable using the CPDAG representation of Markov equivalence classes. We uncover obstacles in translating this approach towards the MAG setting, while also highlighting related open research questions in this area.

## 2  PRELIMINARIES

A mixed graph $G = (V, E)$ consists of a set of vertices and a set of edges between pairs of vertices. We consider three different edge types: directed edges $a \rightarrow b$ or $a \leftarrow b$, bidirected edges $a \leftrightarrow b$, and undirected edges $a - b$. Vertices linked by an edge of any type are called *adjacent* or *neighbors*. The *degree* of a vertex is the number of its neighbors, and the maximum degree of a graph is the maximum degree of any of its vertices. We call vertices connected by a bidirected edge *siblings*, and say that $u$ is a *parent* of $v$ if $u \rightarrow v$ (then $v$ is a *child* of $u$). A path $\pi$ between two vertices $v_1$ and $v_p$ in $G$ is a sequence of distinct vertices $\pi = \langle v_1, \dots, v_p \rangle$ with $p \geq 2$ such that each vertex $v_i$ is adjacent to $v_{i+1}$ for $i = 1, \dots, p-1$. A path of the form $v_1 \rightarrow v_2 \rightarrow \dots \rightarrow v_p$ is directed or causal. If there is a directed path from $u$ to $v$, then $u$ is called an *ancestor* of $v$ and $v$ a *descendant* of $u$. For a vertex $v$, the set of all of its ancestors is written as $An_G(v)$. The descendant set $De_G(v)$ is analogously defined. $Dis_G(v)$ is the set of vertices in the same district as $v$, i.e., the ones connected to $v$ via bidirected edges. Also, we denote by $Pa_G(v)$, $Ch_G(v)$, $Ne_G(v)$, $Si_G(v)$ the set of parents, children, neighbors, siblings of $v$ in $G$, respectively.[1] If $G$ is clear from the context, we omit it as subscript. These notations generalize to sets of vertices in the natural way. We denote the subgraph induced by vertex set $S$ as $G[S] = (S, E \cap (S \times S))$. A graph is *acyclic* if there is no directed path from a vertex $u$ to $v$ with $v \rightarrow u$. An acyclic graph with only directed edges is called a DAG. The *skeleton* of $G$ is the graph obtained by replacing every edge with an undirected one. A *v-structure*, also called an *unshielded collider*, is an ordered triple of vertices $(u, c, v)$ that induces the subgraph $u *\!\!\rightarrow c \leftarrow\!\!* v$. The $*$ indicates that any edge mark is possible. A vertex $c$ on a path $\pi$ is called a *collider* if two arrowheads of $\pi$ meet at $c$, e.g. if $\pi$ contains

---

[1] We note that $v \in An_G(v)$, $v \in De_G(v)$ and $v \in Dis_G(v)$. This does, however, not hold for $Pa_G$, $Ch_G$, $Ne_G$ and $Si_G$.

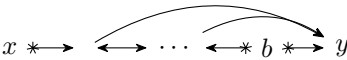

Figure 2: A discriminating path from $x$ to $y$ for $b$. For the last three vertices, $\overset{\frown}{\leftrightarrow} b \leftrightarrow y$ , $\overset{\frown}{\leftrightarrow} b \longrightarrow y$ , and $\overset{\frown}{\longleftarrow} b \longrightarrow y$ are possible configurations (see Fact 4.1). In the first one $b$ is a collider, in the other two a non-collider.

$u \leftrightarrow c \leftarrow v$. Two vertices are *collider connected* if there is a path (a *collider path*) between them on which all internal vertices are colliders; hence, adjacent vertices are collider connected. Vertices are $m$-*connected* by a set $Z$ if there is a path $\pi$ between them on which every collider is in $An(Z)$ and every node that is not a collider is not in $Z$. Such a $\pi$ is called an $m$-*connecting path given $Z$*. If vertices $u, v$ are not $m$-connected by $Z$, written as $(u \perp\!\!\!\perp v \mid Z)_G$, we say that $Z$ $m$-*separates* them. Two sets $X, Y$ are $m$-separated by $Z$ if all their nodes are pairwise $m$-separated by $Z$. In DAGs, $m$-separation is equivalent to $d$-separation [Pearl, 2009].

**Ancestral Graphs.** A graph $G = (V, E)$ is called *ancestral* (AG) if *(i)* it is acyclic, *(ii)* for every bidirected edge $a \leftrightarrow b$ vertex $a$ is not an ancestor of $b$ (and vice versa), and *(iii)* for every undirected edge $a - b$ vertex $a$ (and vertex $b$) have no parents or siblings. Consequently, ancestral graphs contain at most one edge type between two vertices. An AG is a *maximal ancestral graph* (MAG) if set $Z$ exists for every pair of nonadjacent vertices $a$ and $b$ such that $a$ and $b$ are $m$-separated by $Z$. Every AG can be turned into a MAG by adding bidirected edges between vertices that cannot be $m$-separated. Syntactically, all DAGs are MAGs and all AGs that contain only directed edges are DAGs.

**Markov Equivalence.** Two AGs $G_1$ and $G_2$ with the same vertex set $V$ are said to be *Markov equivalent* if we have for all pairwise disjoint sets $A, B, Z \subseteq V$ with $A \neq \emptyset$ and $B \neq \emptyset$ that $A$ and $B$ are $m$-separated given $Z$ in $G_1$ if, and only if, $A$ and $B$ are $m$-separated given $Z$ in $G_2$. The following definition is central for the study of Markov equivalence of MAGs:

**Definition 2.1** ([Richardson and Spirtes, 2002]). *A path $\pi = \langle x, q_1, \ldots, q_p, b, y \rangle$, $p \geq 1$, is called* discriminating *for vertex $b$ in a MAG $G$ if*

*(i) $x$ is not adjacent to $y$ and*
*(ii) any $q_i, 1 \leq i \leq p$, is a collider on $\pi$ and a parent of $y$.*

A discriminating path is illustrated in Fig. 2. For vertices $b$ and $y$ in $G$ denote by $Discr_G(b, y)$ the set of all discriminating paths $\pi = \langle x, q_1, \ldots, q_p, b, y \rangle$ for $b$. Our focus lies on the computational complexity of the following problem:[2]

---

[2]We first deal with the problem for MAGs *without undirected edges*. We later discuss in Section 7 how these can be included with minor modifications (our main theorem holds as is).

**Problem 2.2** (MAG-EQUIVALENCE).

*Instance:*   *Two MAGs $G_1$ and $G_2$.*
*Question:*   *Are $G_1$ and $G_2$ Markov equivalent?*

# 3   HISTORY

A graphical criterion for Markov equivalence of DAGs was provided by Verma and Pearl [1990] and Frydenberg [1990]:

**Theorem 3.1** ([Verma and Pearl, 1990, Frydenberg, 1990]). *Two DAGs $G_1$ and $G_2$ are Markov equivalent if, and only if,*

*(i) $G_1$ and $G_2$ have the same adjacencies and*

*(ii) $G_1$ and $G_2$ have the same unshielded colliders.*

The first graphical criterion for two MAGs to be Markov equivalent was given by Spirtes and Richardson [1996]:

**Theorem 3.2** (Spirtes and Richardson Criterion (SRC)). *Two MAGs $G_1$ and $G_2$ are Markov equivalent if, and only if,*

*(i) $G_1$ and $G_2$ have the same adjacencies,*

*(ii) $G_1$ and $G_2$ have the same unshielded colliders, and*

*(iii) if $\pi$ forms a discriminating path for $b$ in $G_1$ and $G_2$, then $b$ is a collider on the path $\pi$ in $G_1$ if, and only if, it is a collider on the path $\pi$ in $G_2$.*

Note that it is indeed possible that $G_1$ contains a discriminating path for $b$ and $y$, which is not present in $G_2$, even in the case of Markov equivalence (see examples 2 and 3 in Fig. 3). Therefore, testing property *(iii)* naively requires exponential time as one has to consider all discriminating paths for variable $b$ (which may be exponentially many).[3]

On the quest of finding a *polynomial-time*-checkable criterion for the Markov equivalence of MAGs, Zhao et al. [2005] proposed the following characterization:

**Theorem 3.3** ([Zhao et al., 2005]). *Two MAGs $G_1$ and $G_2$ are Markov equivalent if, and only, if $G_1$ and $G_2$ have the same minimal collider paths.*[4]

However, this characteristic also does not lead to a polynomial-time algorithm as there can be exponentially many minimal collider paths. Subsequently, discernible effort has been made to develop an algorithm that tests whether two MAGs are Markov equivalent and that runs in polynomial time [Ali et al., 2009, Hu and Evans, 2020]. To achieve this, the natural formulation in the style of Theorem 3.2 has been abandoned and more involved criteria without an intuitive graphical interpretation were introduced.

---

[3]Spirtes and Richardson [1996] claimed that the criterion is testable in time $n^{O(1)}$, which was later withdrawn [Ali et al., 2009].

[4]$\pi = \langle v_1, \ldots, v_p \rangle$ is minimal if there is no order preserving subsequence $\langle v_1 = v_{i_1}, \ldots, v_t = v_{i_t} \rangle$ that forms a collider path.

Ali et al. [2009] used *triples with order* (if the triple forms a collider, it is called a *collider with order*). The idea behind this approach is to consider only the discriminating paths that are present in any Markov equivalent MAG. While this was an important contribution towards characterizing Markov equivalence classes of MAGs [Ali et al., 2005], the recursive definition of such triples lacks the graphical intuitiveness of, e. g, the SRC. With significant technical effort, the following criterion was developed:

**Theorem 3.4** (Theorem 3.7 in [Ali et al., 2009]). *Two MAGs $G_1$ and $G_2$ are Markov equivalent if, and only if, they have the same adjacencies and the same colliders with order.*

This criterion led to the sought polynomial-time algorithm. However, the dependency is $O(n \cdot m^4)$ for MAGs with $n$ vertices and $m$ edges.

Another criterion was proposed by Hu and Evans [2020] based on so-called *parametrizing sets*. As we compare our algorithm with this approach, we give a brief overview. For a vertex set $W \subseteq V$, the *barren subset* of $W$ is defined as $\mathrm{barren}(W) = \{w \in W \mid De(w) \cap W = \{w\}\}$. A set $H$ is called a *head* if $\mathrm{barren}(H) = H$ and $H$ is contained in a single district in $G[An(H)]$. Let $\mathcal{H}(G)$ be the set of heads and define the *tail* of a head as:

$$\mathrm{tail}(H) = (Dis_{G[An(H)]}(H) \setminus H) \cup Pa_G(Dis_{G[An(H)]}(H)).$$

The parametrizing set of MAG $G$ is defined as the set $\mathcal{S}(G) = \{H \cup A \mid H \in \mathcal{H}(G) \text{ and } A \subseteq \mathrm{tail}(H)\}$. Hu and Evans [2020] showed that MAGs $G_1$ and $G_2$ are Markov equivalent if, and only if, they have the same parametrizing sets. However, generating these sets is costly as they may have exponential size. Hence, they consider $\tilde{\mathcal{S}}_3 \subseteq \mathcal{S}$, which only includes sets $S$ of cardinality 2 and 3, with the vertices in $S$ having 1 or 2 adjacencies.

**Theorem 3.5** (Corollary 3.2.1 in Hu and Evans [2020]). *Two MAGs $G_1$ and $G_2$ are Markov equivalent if, and only if, $\tilde{\mathcal{S}}_3(G_1) = \tilde{\mathcal{S}}_3(G_2)$.*

The sets $\tilde{\mathcal{S}}_3(G)$ can be generated in time $O(nm^2)$, which is significantly faster than the algorithm by Ali et al. [2009]. However, the criterion in this form is quite technical and does not lend itself easily to graphical characterizations of Markov equivalent MAGs.

# 4 A SIMPLE CRITERION FOR THE MARKOV EQUIVALENCE OF MAGS

We propose a *constructive* variant of the Spirtes and Richardson Criterion (SRC) for the Markov equivalence of MAGs. This allows us to develop an efficient equivalence test, improving upon the previous $O(n^5)$ runtime by Hu and Evans [2020]. Additionally, our criterion has a natural graphical interpretation, which is of independent value. We begin with the following fact observed before in Fig. 2.

**Fact 4.1.** *Let $\pi = \langle x, \ldots, q, b, y \rangle$ be a discriminating path in a MAG $G$. Then $b$ and $y$ are connected either via $b \leftrightarrow y$ or $b \to y$ and in the former case $b$ is a collider on $\pi$, in the latter a non-collider.*

*Proof.* Recall that $q$ is collider and a parent of $y$ and, thus, the edge $q \to y$ is present and the edge between $q$ and $b$ is either $q \leftarrow b$ or $q \leftrightarrow b$. To prove the claim, we first show that $b \leftarrow y$ cannot occur and distinguish the two ways the edge between $q$ and $b$ is oriented. If $q \leftarrow b$, then we have a directed cycle $q \to y \to b \to q$; if $q \leftrightarrow b$ we would have $q$ as an ancestor of $b$, which violates the ancestrality property.

For the second part, note that $b$ is always a non-collider if $b \to y$. In case of $b \leftrightarrow y$, the edge $q \leftarrow b$ cannot occur as $b$ would be an ancestor of $y$, violating the ancestrality property. Hence, $b$ is a collider in this case. □

**Theorem 4.2** (Constructive-SRC). *Two MAGs $G_1$ and $G_2$ are Markov equivalent if, and only if,*

*(I) $G_1$ and $G_2$ have the same adjacencies,*

*(II) $G_1$ and $G_2$ have the same unshielded colliders, and*

*(III) for all edges $b \leftrightarrow y \in G_1$ with $Discr_{G_1}(b,y) \neq \emptyset$ we have $b \to y \notin G_2$ and vice versa.*

*Proof.* We first show that if $G_1$ and $G_2$ fulfill the conditions listed above, then they are Markov equivalent by arguing that in this case SRC is satisfied. The first two conditions are identical. Assume the third one holds for the constructive-SRC. Then there is no discriminating path for $(x, b, y)$ with $b \leftrightarrow y$ in $G_1$ (for the argument we only consider $G_1$ w.l.o.g.) such that $b \to y$ in $G_2$. Hence, it cannot happen that we have a discriminating path $\pi$ in $G_1$ and $G_2$ such that $b$ is a collider in $G_1$ and a non-collider in $G_2$ (this is *(iii)* in the SRC). This is because in that case $G_2$ would have $b \to y$ by Fact 4.1.

For the second direction: Assume $G_1$ and $G_2$ violate one of the three conditions *(I)*, *(II)* or *(III)*. We show that they are not Markov equivalent. By the SRC, this is obvious for *(I)* and *(II)*. Now consider that *(III)* is violated but *(I)* and *(II)* are true. Then, w.l.o.g., assume that for some $b \leftrightarrow y$ in $G_1$ there is a discriminating path $\pi = \langle x, q_1, \ldots, q_p, b, y \rangle$ in $G_1$ and the edge $b \to y \in G_2$. It follows by the maximality of $G_1$ and the fact that $x$ and $y$ are nonadjacent that there is a set $Z$ such that $(x \perp\!\!\!\perp y \mid Z)_{G_1}$. One can easily verify that $q_1, \ldots, q_p \in Z$ and $b \notin Z$. Due to the former observation it holds that $(x \not\perp\!\!\!\perp b \mid Z)_{G_1}$. On the other hand, one can see that $(x \perp\!\!\!\perp y \mid Z)_{G_2}$ and $(x \not\perp\!\!\!\perp b \mid Z)_{G_2}$ cannot both hold in $G_2$. This is due to the fact that $(x \not\perp\!\!\!\perp b \mid Z)_{G_2}$ immediately implies $(x \not\perp\!\!\!\perp y \mid Z)_{G_2}$ as $b$ is a non-collider not contained in $Z$. Hence, $G_1$ and $G_2$ are not Markov equivalent. □

To illustrate the constructive-SRC, we give three examples (see Fig. 3) and discuss why or why not Markov equivalence

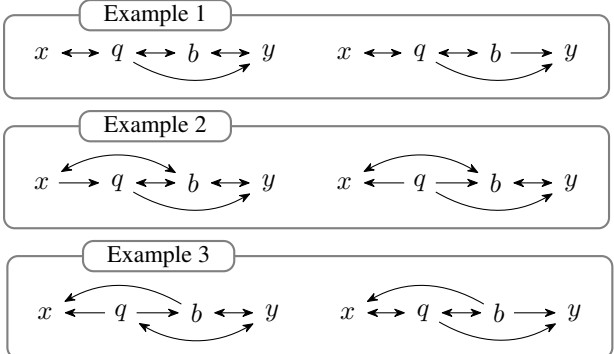

Figure 3: Three examples to illustrate the constructive-SRC. Example 2 is from [Ali et al., 2009] and Example 3 is a modification of another example therein.

holds (as all considered pairs of graphs have the same adjacencies and unshielded colliders, we focus on whether *(iii)* of the SRC and *(III)* of the constructive-SRC are satisfied).

**Example 1** (Fig. 3). The graphs are not Markov equivalent as the left one contains a discriminating path from $x$ to $y$ with $b \leftrightarrow y$ and the right graph contains the edge $b \rightarrow y$, which violates condition *(III)* of the constructive-SRC. In the SRC, condition *(iii)* is not satisfied as the discriminating path $\pi = \langle x, q, b, y \rangle$ exists in both graphs with $b$ being a collider in the left one and a non-collider in the right one.

**Example 2** (Fig. 3). The graphs are Markov equivalent. There is a discriminating path $\langle x, q, b, y \rangle$ in the left graph and it includes the edge $b \leftrightarrow y$, but the right graph also contains the edge $b \leftrightarrow y$ and, hence, *(III)* does not apply. Accordingly, condition *(iii)* of the SRC does not apply as $\langle x, q, b, y \rangle$ is not a discriminating path. An advantage of the constructive-SRC is that one does not have to check for every discriminating path whether it exists in both graphs. It is sufficient to check for the existence of such a path with collider $b$ in one graph, in combination with the edge $b \rightarrow y$ in the other graph.

**Example 3** (Fig. 3). The graphs are Markov equivalent as well. There is no discriminating path in the left graph, but one in the right graph, namely $\langle x, q, b, y \rangle$. It contains $b \rightarrow y$ and, hence, *(III)* does not apply (here, the discriminating path needs to contain $b \leftrightarrow y$ and the *other* graph needs to contain $b \rightarrow y$). Also, *(iii)* does not apply because, as stated above, there is no discriminating path in the left graph.

This third example is interesting, because it highlights that *(III)* indeed only refers to discriminating paths with $b \leftrightarrow y$. If then $b \rightarrow y$ in the other graph, one can conclude that Markov equivalence does not hold. If we have a discriminating path with $b \rightarrow y$, even if $b \leftrightarrow y$ in the other graph, we cannot conclude the same. However, as we have seen above, condition *(III)* is not only necessary for Markov equivalence,

it is, together with *(I)* and *(II)*, also sufficient. This is because *(iii)* in the SRC could only be violated if we have a discriminating path with a collider in one graph (hence $b \leftrightarrow y$) and a non-collider in the other (hence $b \rightarrow y$) and, consequently, *(III)* would be violated as well. Hence, for the constructive-SRC, it is not necessary to consider discriminating paths with non-colliders $b$. This entails a simplification, which makes *(III)* easier to check by hand compared to previous formulations (we discuss the algorithmic advantages of the constructive-SRC in the subsequent section) as one only has to look for discriminating path with collider $b$. Moreover, it also allows to simplify the notion of a discriminating path as *a collider path between non-adjacent $x$ and $y$ for which every vertex but the one before $y$ is a parent of $y$.*

We note that *(III)* is a generalization of the unshielded collider condition *(ii)*. To see this, we reformulate the criterion for Markov equivalence of DAGs (Theorem 3.1):[5]

**Corollary 4.3** ([Verma and Pearl, 1990, Frydenberg, 1990]). *Two DAGs $G_1$ and $G_2$ are Markov equivalent if, and only if,*

*(a)* $G_1$ *and* $G_2$ *have the same adjacencies and*

*(b)* *if in* $G_1$ *there is an unshielded collider* $x \rightarrow b \leftarrow y$, *then* $G_2$ *does not contain* $b \rightarrow y$ *and vice versa.*

*Proof.* We argue that *(b)* is true if, and only if, $G_1$ and $G_2$ have the same unshielded colliders (implying that Corollary 4.3 is equivalent to Theorem 3.1). The first direction is immediate: if one graph contains the unshielded collider $x \rightarrow b \leftarrow y$ while the other graph orients $b \rightarrow y$, then clearly $(x, b, y)$ is an unshielded collider in only one them.

For the other direction assume w.l.o.g. that $G_1$ contains an unshielded collider $(u, v, w)$, but $G_2$ does not. Then $G_2$ has either $u \leftarrow v$ or $v \rightarrow w$. In both cases *(b)* is violated (set $b = v$ and either $x = w, y = u$ or $x = u, y = w$). □

**Corollary 4.4.** *Two MAGs $G_1$ and $G_2$ are Markov equivalent if, and only if,*

*(A)* $G_1$ *and* $G_2$ *have the same adjacencies,*

*(B)* *if there is a collider path* $\langle x, \ldots, b, y \rangle$ *between non-adjacent $x$ and $y$ with every vertex but $x$, $b$ and $y$ being a parent of $y$ in $G_1$, then $G_2$ does not contain the edge* $b \rightarrow y$ *and vice versa.*

*Proof.* The collider path $\langle x, \ldots, b, y \rangle$ may only consist of three vertices, i. e., it could be an unshielded collider. If the other graph were to contain the edge $b \rightarrow y$, then it would not have that same collider, meaning the graphs are not Markov equivalent by *(II)*. If the collider path consists of more than three vertices, the formulation equals *(III)*. □

---

[5]There are even further formulations of *(III)*, e. g., in terms of parameterizing sets, as pointed out by an reviewer: If there is a discriminating path for $\{x, b, y\}$ with non-collider $b$, then the set is parameterizing in both graphs.

We remark that this corollary applies only to MAGs without undirected edges (in contrast to the constructive-SRC). However, only minor modifications are necessary to handle undirected edges as well. We discuss these in Section 7.

# 5 TESTING MARKOV EQUIVALENCE OF MAGS ALGORITHMICALLY

In the previous section, we derived a simple characterization of Markov equivalence for MAGs. In this section, we deal with the computational side of the problem and discuss how this new characterization can be tested. The algorithm we propose has a worst-case runtime of $O(n^3)$, thus being significantly faster than previous approaches. Moreover, for sparse graphs, which are very common in causal modeling, we even report linear time in the number of vertices.

We check the conditions *(I)* and *(II)* naively. For checking the third condition *(III)*, we need to test for each $b \leftrightarrow y$ in $G_k$ with $k \in \{1,2\}$, for which $b \rightarrow y$ is an edge in the other graph $G_{k'}$ (with $k' = 3 - k$), whether there is a discriminating path for $b$ and $y$. We do this by considering every choice of $y$ consecutively, computing for each the bidirected connected components of its parents (we call these the *parent districts*) that support our computations.

**Definition 5.1.** *Given a MAG $G = (V, E)$ and a vertex $y$, the bidirected connected components of $G[Pa(y)]$ are termed the parent districts of $y$ and denoted as $\mathcal{D}(y)$.*

This notion is useful as the middle part of a discriminating path consists solely of such vertices $q_1, \ldots, q_p$ in a single parent district of $y$. Once the parent districts have been computed, one can check if, for a certain district $D \in \mathcal{D}(y)$, there is a vertex $x$ non-adjacent to $y$ and a parent or sibling of $D$, which can function as the start of the discriminating path. If this is the case, it remains to consider all vertices $b$ which are siblings of $D$ and $y$. For these, we can conclude that they are part of a discriminating path $x \ast\!\!\rightarrow q_1 \leftrightarrow \ldots \leftrightarrow q_p \leftrightarrow b \leftrightarrow y$. If $b \rightarrow y$ in the other graph, the graphs are not Markov equivalent. Figure 4 illustrates this approach and Algorithm 1 gives an implementation.

**Theorem 5.2.** *Algorithm 1 checks whether two MAGs are Markov equivalent in time $O(n^3)$ for general graphs and expected time $O(n \cdot \Delta^2)$ for graphs with maximal degree $\Delta$.*

*Proof.* For the correctness of Algorithm 1, we need to show that *(III)* of the constructive-SRC is correctly checked. If the algorithm returns *Not Markov equivalent* in line 13, then there exists a $b$ and $y$ such that $b \leftrightarrow y$ in one graph and $b \rightarrow y$ in the other. Moreover, in the former graph there exists a parent district $D \in \mathcal{D}(y)$ such that there is a $x \in Pa_{G_k}(D) \cup Si_{G_k}(D) \setminus Ne_{G_k}(y)$ (this set is non-empty in line 8) and it is guaranteed that $b$ is not only a sibling of $y$, but also of $D$. Hence, there is a discriminating path

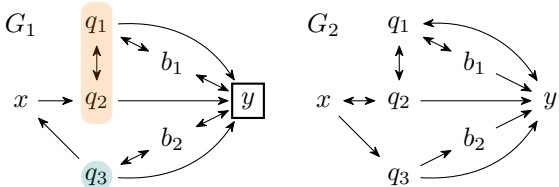

Figure 4: Algorithm 1 checking vertex $\boxed{y}$ in $G_1$ (line 5) with $\mathcal{D}(y) = \{\ \{q_3\}\ ,\ \{q_1, q_2\}\ \}$. For $D = \{q_3\}$, the set $Pa_{G_1}(D) \cup Si_{G_1}(D) \setminus Ne_{G_1}(y)$ is empty as $x$ is a child of $D$. Hence, Algorithm 1 does not consider $D$ further (line 8). For $D = \{q_1, q_2\}$, the set $Pa_{G_1}(D) \cup Si_{G_1}(D) \setminus Ne_{G_1}(y)$ contains $x$, which is a parent of $D$ but not a neighbor of $y$. Moreover, $b_1$ is a sibling of both $D$ and $y$. Hence, we obtain the discriminating path $x \rightarrow q_2 \leftrightarrow q_1 \leftrightarrow b_1 \leftrightarrow y$. As $b_1 \rightarrow y$ in $G_2$, the algorithm reports that the graphs are not Markov equivalent. Note that for SRC *(iii)* is violated due to the discriminating path $x, q_2, q_1, y$ in both graphs with $G_1$ containing non-collider $q_2 \leftrightarrow q_1 \rightarrow y$ and $G_2$ containing collider $q_2 \leftrightarrow q_1 \leftrightarrow y$. The discriminating path for $q_1 \leftrightarrow y$ in $G_2$ and the corresponding edge $q_1 \rightarrow y$ would also be detected by Algorithm 1. Note that here $\{q_2\}$ is a parent district of $y$ ($q_1$ is not part of this district as it is not a parent of $y$ in $G_2$).

$x \ast\!\!\rightarrow q_1 \leftrightarrow \ldots \leftrightarrow q_p \leftrightarrow b \leftrightarrow y$, with $q_1, \ldots, q_p \in D$ and $q_1$ being the sibling/child of $x$ and $q_p$ being the sibling of $b$. The collider path from $q_1$ to $q_p$ exists by the definition of $D$. For the same reason, we have that $q_1, \ldots, q_p$ are parents of $y$. Note, in particular, that $b \neq x$ and both are not in $D$. Thus, *(III)* is violated and the output is correct.

For the other direction, if the graph contains a violation of *(III)*, then there is a discriminating path for $b \leftrightarrow y$ (while the other graph contains $b \rightarrow y$). The existence of such a path is detected as all discriminating paths for $b \leftrightarrow y$ have the form $x \ast\!\!\rightarrow q_1 \leftrightarrow \ldots \leftrightarrow q_p \leftrightarrow b \leftrightarrow y$ with $q_1, \ldots, q_p$ being parents of $y$. Thus, there is a parent district of $y$, which has $x$ as parent/sibling and $b$ as a sibling. Hence, the algorithm outputs *Not Markov equivalent* in line 13.

Regarding the runtime, note that checking *(I)* and *(II)* in line 1 is possible in time $O(n^2)$, resp. $O(n^3)$. If the graph is sparse the runtimes $O(n\Delta)$, resp. $O(n\Delta^2)$, follow (for the latter case consider for each vertex all pairs of its parents and test whether they are adjacent.[6]) For checking *(III)*, there are $n$ vertices $y$ considered per graph at line 5. Computing the parent districts for one $y$ can be done in time $O(n^2)$ or $O(\Delta^2)$ if $\Delta$ is the maximal degree of the graph, as finding the connected components of a (sub)graph with $s$ vertices takes $O(s^2)$ time in the worst-case. Hence, this step can be performed in time $O(n^3)$ or $O(n \cdot \Delta^2)$. Also the neighbors/parents/siblings of $y$ and all its parent districts

---

[6] We can perform adjacency tests in $O(1)$ by storing the graph as adjacency matrix. For sparse graphs we may avoid $O(n^2)$ space by using hash tables, which yields expected time $O(1)$.

may be precomputed in this phase as well.

Further, there are two nested for loops, one over the parent districts (there are at most $O(n)$ or $O(\Delta)$ many) and one over $b$ (again there are $O(n)$ or $O(\Delta)$ choices for $b$)[7]. Finally line 11 can be performed in (expected) time $O(1)$ (see footnote 6), yielding again $O(n \cdot \Delta^2)$ proving the claim. $\square$

---

**input** : Two MAGs $G_1 = (V_1, E_1)$, $G_2 = (V_2, E_2)$.
**output** : Whether $G_1$ and $G_2$ are Markov equivalent.

1 **if** *(I) or (II) of the constructive-SRC is violated* **then**
2     **return** Not Markov equivalent.
3 **end**
4 **foreach** $G_k = (V_k, E_k)$ *with* $k \in \{1, 2\}$ **do**
5     **foreach** $y \in V_k$ **do**
6        **foreach** $D \in \mathcal{D}(y)$ **do**
7           Compute $Pa_{G_k}(D)$ and $Si_{G_k}(D)$.
8           **if** $Pa_{G_k}(D) \cup Si_{G_k}(D) \setminus Ne_{G_k}(y) \neq \emptyset$ **then**
9              **foreach** $b \in Si_{G_k}(D) \cap Si_{G_k}(y)$ **do**
10                 Let $G_{k'}$ be the other graph, i. e., $k' = 3 - k$.
11                 **if** $b \rightarrow y$ *in* $G_{k'}$ **then**
12                    **return** Not Markov equivalent.
13                 **end**
14              **end**
15           **end**
16        **end**
17     **end**
18 **end**
19 **return** Markov equivalent.

**Algorithm 1:** Checking the constructive-SRC.

---

We conclude that for graphs with maximal degree $\Delta$ the expected runtime can be written as $O(n \cdot \Delta^2)$, which is linear in the number of vertices for a constant $\Delta$. We note that this is a significant improvement over Hu and Evans [2020], who reported time $O(m^2) = O(n^2)$ for sparse random graphs.

# 6   A DIFFERENT APPROACH TO MARKOV EQUIVALENCE TESTING

The continued improvement of algorithms for testing the Markov equivalence of MAGs from exponential time (SRC) over $O(n^9)$ (Ali et al. [2009]) and $O(n^5)$ (Hu and Evans [2020]) to $O(n^3)$ begs the question of what the best achievable runtime is. Is it possible to test Markov equivalence of MAGs in $O(n^2)$? A natural comparison is the one to the Markov equivalence of DAGs. Here, the *naïve* test of Theorem 3.1 can be done in $O(n^3)$: List all triples that are unshielded colliders. This approach cannot lead to faster algorithms, as we may have $\Omega(n^3)$ unshielded colliders (this

obstacle exists for MAGs as well). This indicates that a whole new approach is necessary.

For DAGs, such an approach is possible by utilizing *completed partially directed acyclic graph* (CPDAGs) [Andersson et al., 1997]. A CPDAG is a compact and unique representation of a Markov equivalence class. To test whether two DAGs are Markov equivalent, one may compute the corresponding CPDAGs $C_1$ and $C_2$ and check whether $C_1 = C_2$. The complexity of this approach hinges on the complexity of converting DAGs to CPDAGs. There are two algorithmic strategies for this task: The first one imitates the PC algorithm for learning the CPDAG from observational data [Spirtes et al., 2000]. First, initialize $C$ as the skeleton of $D$. Second, set all v-structures of $D$ in $C$. Third, orient further edges by repeated application of the first three Meek rules [Meek, 1995]. The second strategy constructs the CPDAG from $D$ based on a topological ordering of $D$ while utilizing characterizations of CPDAGs and Markov equivalence classes of DAGs [Andersson et al., 1997]. This approach was used by [Chickering, 1995], who proposed a clever linear-time (i. e., $O(n + m)$) algorithm for the DAG-to-CPDAG task.

Hence, based on the second approach, testing Markov equivalence of DAGs can be done in linear time $O(n + m)$.[8]

Coming back to MAGs, we note that the first approach for DAGs can be used as well. For a MAG $G$, one can imitate the FCI algorithm [Spirtes et al., 2000], which is the counterpart of the PC algorithm under latent confounding/selection bias, to obtain its corresponding *partial ancestral graph* (PAG) [Zhang, 2008a], which is, analogously to the CPDAG for DAGs, a compact and unique representation of an equivalence class. This is done by first initializing $P$ as the skeleton of $G$, setting the unshielded colliders according to $G$ and, finally, applying the 10 completion rules given by Zhang [2008b] (see also Ali et al. [2005]). This approach yields a polynomial-time algorithm for testing Markov equivalence of MAGs, but with a rather large polynomial:[9] One can compute the PAGs $P_1$, $P_2$ for the given MAGs $G_1$, $G_2$ and check whether they are identical[10].

The second strategy currently cannot be translated to MAGs as there is no counterpart for the DAG-to-CPDAG algorithm to directly transform a MAG into a PAG. Hence, a better understanding of PAGs might be needed for further progress and we deem this as an important topic for future research.

---

[7]Forming $Si_{G_k}(D) \cap Si_{G_k}(y)$ can be done in $O(n)$. To do it in expected time $O(\Delta)$ we may again use hash tables.

[8]The runtime of the first approach depends on the complexity of orienting the graph with the Meek rules. Wienöbst et al. [2021a] showed that it is possible to perform this step in $O(n^3)$.

[9]The time is a polynomial of order roughly $O(m^3 \cdot n)$ as for every undirected edge we have to check whether global conditions hold (Zhang [2008b] briefly discuss the runtime, mentioning $O(n \cdot m)$ for checking the fourth rule for a single edge.

[10]This strategy has some parallels to Ali et al. [2009], due to the fact that colliders with order also play a key role in the completeness of the FCI rules Ali et al. [2005].

| $n$ | Algorithm HE | | Algorithm C-SRC | |
|---|---|---|---|---|
| | Avg. Time | Std. Dev. | Avg. Time | Std. Dev. |
| 250 | 0.0487s | 0.0101 | 0.0015s | 0.0009 |
| 500 | 0.1058s | 0.0388 | 0.0032s | 0.0051 |
| 750 | 0.1605s | 0.0279 | 0.0049s | 0.0065 |
| 1000 | 0.2587s | 0.0594 | 0.0062s | 0.0058 |
| 1250 | 0.3579s | 0.0684 | 0.0085s | 0.0081 |
| 1500 | 0.4629s | 0.0789 | 0.0091s | 0.0058 |
| 1750 | 0.5373s | 0.0626 | 0.0106s | 0.0021 |
| 2000 | 0.6794s | 0.0778 | 0.0119s | 0.0024 |

*Random graphs with $n =$*

250  500  750  1000  1250  1500  1750  2000

Advantage of C-SRC / HE

0.0472s  0.10258s  0.15556s  0.2525s  0.34938s  0.45377s  0.52664s  0.66751s

| $n$ | Algorithm HE | | Algorithm C-SRC | |
|---|---|---|---|---|
| | Avg. Time | Std. Dev. | Avg. Time | Std. Dev. |
| 25 | 0.0011s | 0.0007 | 0.0004s | 0.0004 |
| 50 | 0.0169s | 0.0028 | 0.0028s | 0.0007 |
| 75 | 0.0912s | 0.0246 | 0.0092s | 0.0014 |
| 100 | 0.4004s | 0.1037 | 0.0263s | 0.0129 |
| 125 | 1.0339s | 0.2283 | 0.0466s | 0.0091 |
| 150 | 2.3356s | 0.4349 | 0.0808s | 0.0084 |
| 175 | 4.7182s | 0.8741 | 0.1303s | 0.0106 |
| 200 | 8.9285s | 1.5242 | 0.2033s | 0.0129 |

*Random graphs with $n =$*

25  50  75  100  125  150  175  200

Advantage of C-SRC / HE

0.00067s  0.01414s  0.0826s  0.37418s  0.98724s  2.25485s  4.58789s  8.72528s

Figure 5: *Advantage plots* that compare our implementation (C-SRC) with the algorithm by Hu and Evans (HE). Each bar corresponds to an experiment on random graphs with $n$ vertices (denoted above the bars) and $k = 3n$ (top image) or $k = 10n$ (bottom image) edges, respectively. For each experiment we measured the average time needed by both algorithms over 250 instances. If C-SRC uses $t_1$ seconds and HE took $t_2$ seconds, then the *advantage* of C-SRC over HE is defined by $t_2 - t_1$ (i.e., the advantage is positive iff C-SRC is faster). The advantage (in seconds) is shown below the bars.

## 7 RELATED PROBLEMS

So far, our focus lied on the problem of testing Markov equivalence of MAGs without undirected edges. In this section we discuss the connection to more general formulations of the problem. First, we note that the constructive-SRC and Algorithm 1 also work for MAGs with undirected edges. This is because the SRC also holds in this setting and that there cannot be an undirected edge in a discriminating path (in particular, the edge between $b$ and $y$ cannot be undirected). For Corollary 4.4 a modification is necessary: condition *(II)* has to be changed to "If there is a collider path $x, \dots, b, y$ between non-adjacent $x$ and $y$ with every vertex but $x$, $b$ and $y$ being a parent of $y$ in one graph, then the other graph does neither contain the edge $b \rightarrow y$ *nor* the edge $b - y$." This is necessary as a collider $u \leftrightarrow v \leftrightarrow w$ in one graph might correspond to a non-collider $u - v - w$ in the other graph – and these graphs are, of course, not Markov equivalent.

Further related problems are obtained by removing the maximality or the ancestrality requirement (or both). In that case, we deal with general *acyclic directed mixed graphs* (ADMGs). These are graphs that may contain directed and bidirected edges with the only requirement that there is no directed cycle. The SRC and constructive-SRC do not apply for ADMGs as they explicitly use the maximality and ancestrality properties. However, one can transform ADMGs into equivalent MAGs and, thus, test the Markov equivalence of ADMGs using the algorithms for MAGs. As it turns out, the currently fasted algorithm for the ADMG-to-MAG transformation (Algorithm 2 in Hu and Evans [2020]) requires time $O(n^4)$ and is, thus, the bottleneck in this approach (testing the equivalence of MAGs is in $O(n^3)$ by Theorem 5.2).

It is unclear to us whether this transformation can be performed in $O(n^3)$. A central part of it involves the computation of so-called *inducing paths*, where it has to be checked for every pair of vertices $(x, y)$ whether there is a collider path between $x$ and $y$ via vertices in $An(x, y)$. Since we have $O(n^2)$ such pairs and since, further, graph traversal is in $\Omega(n + m)$, this direct approach necessarily produces a workload of $O(m \cdot n^2)$. We believe that it will be central for the developement of faster ADMGs equivalence tests to better understand the complexity of ADMG-to-MAG and consider this as an interesting question for further work.

Table 1: Distribution of directed edges (→) and bidirected edges (↔) in the randomly generated ADMGs and in the corresponding MAGs. For every row we generated 250 random ADMGs with $n$ vertices and $k$ edges, and show the average of directed or bidirected edges they contain.

| | | ADMG | | | MAG | | |
|---|---|---|---|---|---|---|---|
| $n$ | $k$ | → | ↔ | →/↔ | → | ↔ | →/↔ |
| 250 | $3n$ | 373.844 | 376.156 | 0.9962 | 394.38 | 401.88 | 0.9840 |
| 500 | $3n$ | 752.536 | 747.464 | 1.0081 | 772.148 | 771.456 | 1.0022 |
| 750 | $3n$ | 1125.812 | 1124.188 | 1.0023 | 1146.012 | 1147.944 | 0.9991 |
| 1000 | $3n$ | 1500.308 | 1499.692 | 1.0010 | 1519.616 | 1523.628 | 0.9980 |
| 1250 | $3n$ | 1873.564 | 1876.436 | 0.9990 | 1892.58 | 1899.968 | 0.9966 |
| 1500 | $3n$ | 2251.344 | 2248.656 | 1.0016 | 2270.228 | 2272.872 | 0.9992 |
| 1750 | $3n$ | 2627.564 | 2622.436 | 1.0023 | 2647.156 | 2646.568 | 1.0005 |
| 2000 | $3n$ | 3002.328 | 2997.672 | 1.0018 | 3021.66 | 3021.5 | 1.0003 |
| 25 | $10n$ | 124.392 | 125.608 | 0.9984 | 246.432 | 49.544 | 5.4013 |
| 50 | $10n$ | 250.912 | 249.088 | 1.0113 | 824.804 | 306.212 | 2.7741 |
| 75 | $10n$ | 374.312 | 375.688 | 0.9989 | 1639.868 | 796.848 | 2.0995 |
| 100 | $10n$ | 498.536 | 501.464 | 0.9962 | 2684.468 | 1501.252 | 1.8099 |
| 125 | $10n$ | 625.692 | 624.308 | 1.0038 | 3902.696 | 2456.212 | 1.6067 |
| 150 | $10n$ | 749.684 | 750.316 | 1.0006 | 5314.2 | 3632.024 | 1.4762 |
| 175 | $10n$ | 874.788 | 875.212 | 1.0007 | 6926.444 | 4999.7 | 1.3952 |
| 200 | $10n$ | 1002.088 | 997.912 | 1.0051 | 8662.34 | 6582.824 | 1.3236 |

## 8 EXPERIMENTS

To emphasize the practical effectiveness of the constructive-SRC and, in particular, Algorithm 1, we compare it experimentally with the algorithm proposed by Hu and Evans [2020] on synthetic data. Both algorithms were implemented in the Julia programming language [Bezanson et al., 2017] and we ran the experiments on a desktop computer with an Intel(R) Core(TM) i7-8565U CPU and 16GBs of RAM.[11] Synthetic MAGs were generated with the process described in [Hu and Evans, 2020]: Fix a topological ordering $\tau$ of the vertices, then add $k$ edges uniformly at random, and finally direct each edge with probability $1/2$ according to $\tau$. Replacing remaining undirected edges with bidirected edges yields an ADMG, which can in turn be transformed into a MAG, as discussed in Section 7.

For a fair comparison with the experiments in [Hu and Evans, 2020], we run a modified version of Algorithm 1. It generates, for *a single MAG*, a set of all adjacencies $A$ (for checking *I*), a set of all v-structures $V$ (for checking *II*), and the set $C$ of all $b \leftrightarrow y$ that are part of a discriminating path, as well as the set $N$ of all $b \rightarrow y$ (for checking *III*). Clearly, if one were to generate these sets for two MAGs $G_1$ and $G_2$, testing equivalence would reduce to checking whether $A_1 = A_2$, $V_1 = V_2$, $C_1 \cap N_2 = \emptyset$, and $C_2 \cap N_1 = \emptyset$.

This approach provides a finer control over the experiments as it avoids the possibility of an "early stopping" at line 1 or line 13 of Algorithm 1 (which can happen if the given MAGs are not equivalent and would give an unfair advantage to our algorithm). It also enables us to consider single MAGs, which are simpler to generate randomly than, e. g, two random Markov equivalent MAGs. The reported run-

times can be viewed, for our algorithm as well as for [Hu and Evans, 2020], as essentially half the time occurring when two Markov equivalent DAGs are compared (because these steps have to be performed for both graphs).

For the choice of the parameter $k$ (the number of edges), we follow, on the one hand, Hu and Evans [2020] and set it to $k = 3n$ (see the top part of Fig. 5) and, on the other hand, also consider denser graphs with $k = 10n$ (bottom part of Fig. 5). Note that $k$ is the number of edges in the generated ADMG and not in the MAGs on which the algorithms run. The transformation of an ADMG into a MAG might generate new edges, see Table 1. For $k = 3n$, one usually only sees a small increase, while a significant amount of edges is added for $k = 10n$. The proportion of directed and bidirected edges also changes in the latter case, the graphs usually contain more directed edges than bidirected ones. For our experiments, we ran both algorithms on the same 250 randomly generated graphs for each choice of parameters and report the average time they used in Fig. 5.

It can be seen that Algorithm 1 is faster for all choices of $n$ and $k$. We can also observe that for ever larger graphs, the advantage increases – which implies that the algorithm in fact has a better asymptotic behaviour. This phenomenon becomes even more significant on the dense (and, thus, more difficult) instances. Finally, the absolute runtime of Algorithm 1 is generally extremely low (for the considered inputs only fractions of a second[12]).

## 9 CONCLUSIONS

We proposed the constructive-SRC – a new criterion for the Markov equivalence of MAGs. It is expressed in terms of natural graphical concepts, can easily be tested by hand for smaller graphs, and leads to the first cubic-time algorithm.

For further work, it remains an open problem whether the runtime can be reduced to $O(n^2)$, as is possible for DAGs. We argued that a different approach is necessary, as any approach that explicitly considers all unshielded colliders has a complexity of $\Omega(n^3)$. Generally, a better understanding of Markov equivalence classes of MAGs may facilitate the translation of further research from the DAG setting, e. g., regarding active learning [Hauser and Bühlmann, 2012] or the question of computing the size of Markov equivalence classes [Wienöbst et al., 2021b], which could add to recent results in this direction [Kocaoglu et al., 2019, Wang and Zhou, 2021]. Finally, due to the improved runtime for equivalence testing of MAGs, the ADMG-to-MAG transformation is currently the bottleneck for the problem on acyclic mixed graphs, making the design of a faster transformation algorithm an important task for further work.

---

[11]The code is available under: `https://github.com/mwien/magequivalence`

[12]Generating significantly harder instances is not a trivial task as the random generation process relies on the ADMG-to-MAG task, which currently cannot be performed faster than in $O(n^4)$.

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
