# OpenReview forum: "A New Constructive Criterion for Markov Equivalence of MAGs"
_auai.org/UAI/2022/Conference — UAI 2022 Oral_

### Official Review · Reviewer_4f65 · 2022-04-10

**Q2(1) Originality/Novelty:** 3
**Q2(2) Significance/Impact:** 2
**Q2(3) Correctness/Technical Quality:** 3
**Q2(6) Clarity Of Writing:** 3
**Q6 Overall Score:** 6
**Q8 Confidence In Your Score:** 3

**Q1 Summary And Contributions:**

Paper propose a graphical criterion for the Markov equivalence of MAGs, which results in an algorithm  with time complexity  of  O(n^3) to check the Markov equivalence of two MAGs.

**Q2 Assessment Of The Paper:**

More detailed information regarding each of these aspects is given below:

**Q2(4) Quality Of Experiments (Optional):**

3: Good: The experimental evaluation is adequate, and the results convincingly support the main claims.

**Q2(5) Reproducibility:**

3: Good: Key resources (e.g., proofs, code, data) are available and key details (e.g., proofs, experimental setup) are sufficiently well-described for competent researchers to confidently reproduce the main results.

**Q3 Main Strengths:**

The state-of-the-art algorithm, provided by Hu and Evans [UAI 2020], runs in time  O(n^5)  with  n vertices for the same task of this paper, while the algorithm proposed in this paper reach O(n^3).

**Q4 Main Weakness:**

 MAGs can not be learned from data directly,  it is difficult for this improvement to bring substantial benefits for causal learning.


**Q5 Detailed Comments To The Authors:**

The new equivalent condition is meaningful theoretically but  the contribution is small. More discussion about potential  applications  are welcomed.

**Q7 Justification For Your Score:**

The new equivalent condition is meaningful theoretically

**Q9 Complying With Reviewing Instructions:**

1: Yes.

---

### Official Review · Reviewer_nWQh · 2022-04-11

**Q2(1) Originality/Novelty:** 3
**Q2(2) Significance/Impact:** 3
**Q2(3) Correctness/Technical Quality:** 4
**Q2(6) Clarity Of Writing:** 4
**Q6 Overall Score:** 7
**Q8 Confidence In Your Score:** 4

**Q1 Summary And Contributions:**

The paper proposes a new characterization for the Markov equivalence of MAGs that is based on the original characterization by (Spirtes and Richardson, 1996). The new characterization enables testing for the equivalence of two MAGs in $O(n^3)$ and improves on the state-of-the-art result which was $O(n^5)$ (for dense graphs).

**Q2 Assessment Of The Paper:**

More detailed information regarding each of these aspects is given below:

**Q2(5) Reproducibility:**

3: Good: Key resources (e.g., proofs, code, data) are available and key details (e.g., proofs, experimental setup) are sufficiently well-described for competent researchers to confidently reproduce the main results.

**Q3 Main Strengths:**

The paper is very well-written and self-contained with great clarifying examples. The proposed characterization (Thm. 4.2) is novel and provides a significant computational advantage (in Algorithm 1) compared to the state-of-the-art. This advancement is achieved through a simple, yet effective and critical, observation in Fact 4.1.

**Q4 Main Weakness:**

I don't see any major weaknesses when it comes to the theoretical results. One issue I can point out is that the introduction and related work occupy almost half of the space (3.5 pages) while the experiments section is moved to the appendix. I understand the motive for the authors to provide a smooth and self-contained paper, which I did enjoy reading, but it would be preferable to include the majority of the experiments section in the main body of the paper.

**Q5 Detailed Comments To The Authors:**

Below are some minor issues:

1- Descendant and ancestor sets: The definition of An(v) and De(v) in p.2 does not specify if v is also included in the set. A similar issue applies to the parents, children, ...

2- district: In the literature, this is more widely referred to as c-component. Is there a reason to deviate from that label and use "district"?

3- Example 2. "one does not have to check for every path $\pi$ whether": It took me a bit to recall that $\pi$ is in reference to $\langle x, q, b, y\rangle$ from Example 1. Maybe include the reference again in Example 2.

4- Section 5. "We do this by considering the vertices y consecutively": This statement is puzzling until the reader checks Algorithm 1. Otherwise, I thought that y is a set of vertices.

**Q7 Justification For Your Score:**

The proposed characterization is novel and improves on the state-of-the-art, and the weakness in Q4 does not threaten the validity and significance of the result.

**Q9 Complying With Reviewing Instructions:**

1: Yes.

---

### Official Review · Reviewer_KiDQ · 2022-04-13

**Q2(1) Originality/Novelty:** 3
**Q2(2) Significance/Impact:** 2
**Q2(3) Correctness/Technical Quality:** 4
**Q2(6) Clarity Of Writing:** 4
**Q6 Overall Score:** 8
**Q8 Confidence In Your Score:** 4

**Q1 Summary And Contributions:**

This paper provides an algorithm to determine whether or not two MAGs are Markov equivalent. The procedure is motivated by one of the earlier characterizations of MAG Markov equivalence which checks for equality of adjacencies, unshielded colliders, and colliders at the end of discriminating paths. The main innovation of this work is an efficient check for the latter of the three aforementioned conditions.

**Q2 Assessment Of The Paper:**

More detailed information regarding each of these aspects is given below:

**Q2(4) Quality Of Experiments (Optional):**

3: Good: The experimental evaluation is adequate, and the results convincingly support the main claims.

**Q2(5) Reproducibility:**

4: Excellent: Key resources (e.g., proofs, code, data) are available and key details (e.g., proof sketches, experimental setup) are comprehensively described for competent researchers to confidently and easily reproduce the main results.

**Q3 Main Strengths:**

The algorithm and arguments of this paper are clean, clear, and easy to follow. Theoretically, the authors show that their algorithm runs in O(nd^2) time where d is the maximum degree of the graphs being checked (whereas the previous best algorithm ran in O(ne^2) times where e is the number of edges of the graphs being checked). Empirically, the authors compare their result to the current state-of-the-art and demonstrate an impressive performance boost (both algorithms are implemented in Julia).

**Q4 Main Weakness:**

The main weakness of this work is that it is on a niche topic, however, I do think that UAI is the correct venue for this topic.

**Q5 Detailed Comments To The Authors:**

“However, the criterion is quite involved and verifying its satisfiability with pencil and paper, even for small instances, is not simple.”

I disagree, checking for heads with cardinality 3 or less and identifying their corresponding tails is fairly straightforward to do via a visual inspection (no more difficult than identifying discriminating paths via a visual inspection in my opinion).


“We denote the subgraph induced by vertex set S as G[S]”

Please make clear what the edge set is in the induced subgraph.


“A v-structure, also called an unshielded collider, is an ordered triple of vertices (u, c, v) that induces the subgraph u c v”

Please be clear if an unshielded collider can contain bi-driected edges.


“Vertices are m-connected by a set Z if there is a path π between them on which every collider is in An(Z) and every node that is not a collider is not in Z.”

The collider can also have one of the endpoints as an ancestor.


“However, the criterion in this form is not very intuitive in graphical terms and checking such a criterion by hand, respectively using it for a graphical characterization of Markov equivalent MAGs, appears to be inconvenient.”

See earlier comment regarding heads of cardinality 3 or less.


As a connection to prior work, I suggest noting that Theorem 4.2 (III) can be stated in terms of parameterizing sets: For all {x,b,y} defined by a discriminating path (x *-> q … q <-> b *-> y), the set is parameterizing in both graph.


I suggest writing out the full big oh notation in the proof of Theorem 5.2 before simplifying for clarity. I believe it should be O(n(d^2 + d^2)) which equals O(nd^2).


“MAGs cannot be topologically ordered in a meaningful way as easily as DAGs due to the additional bidirected edges”

The vertices of a MAG can be ordered so that all vertices that follow a vertex in the order are not ancestors of that vertex in the MAG. An order of this nature is said to be consistent with the MAG and is used in results such as the order local Markov property.

Thanks you for your detailed comments. I am happy with the paper and the your responses, accordingly, my score will remain as an 8.

**Q7 Justification For Your Score:**

The paper is easy to recommend as a strong accept because it is well written and quite good technically, and the addressed problem has a history of being published at UAI.

**Q9 Complying With Reviewing Instructions:**

1: Yes.

---

### Official Review · Reviewer_9xjE · 2022-04-13

**Q2(1) Originality/Novelty:** 3
**Q2(2) Significance/Impact:** 2
**Q2(3) Correctness/Technical Quality:** 3
**Q2(6) Clarity Of Writing:** 4
**Q6 Overall Score:** 8
**Q8 Confidence In Your Score:** 3

**Q1 Summary And Contributions:**

The main contribution of the paper is a new criterion for testing whether two maximal ancestral graphs (MAGs) are Markov equivalent running in cubic time.

**Q2 Assessment Of The Paper:**

More detailed information regarding each of these aspects is given below:

**Q2(4) Quality Of Experiments (Optional):**

3: Good: The experimental evaluation is adequate, and the results convincingly support the main claims.

**Q2(5) Reproducibility:**

4: Excellent: Key resources (e.g., proofs, code, data) are available and key details (e.g., proof sketches, experimental setup) are comprehensively described for competent researchers to confidently and easily reproduce the main results.

**Q3 Main Strengths:**

The new proposed algorithm for testing Markov equivalence of MAGs has lower computational complexity than any other currently used algorithm. The computational advantage was verified on also experimentally on randomly generated sparse and dense graphs.

**Q4 Main Weakness:**

Since MAGs are still not used as often as DAGs the current applicability is not as high - but this is not a true weakness of the paper.

**Q5 Detailed Comments To The Authors:**

The paper is well written, the key concepts are explained and illustrated using examples.
The actual algorithm is simple (its pseudocode has only 19 lines) and easy to implement.

Minor point in References:
Use capital M in Markov:
- Zhongyi Hu and Robin Evans. Faster algorithms for Markov equivalence.
- R Ayesha Ali, Thomas S Richardson, Peter Spirtes, and Jiji Zhang. Towards characterizing Markov equivalence classes for directed acyclic graphs with latent variables.
- Alain Hauser and Peter Bühlmann. Characterization and greedy learning of interventional Markov equivalence classes of directed acyclic graphs.
Use capital P and C in PC-algorithm:
Markus Kalisch and Peter Bühlman. Estimating high-dimensional directed acyclic graphs with the PC-algorithm.


**Q7 Justification For Your Score:**

Nice contribution with a new result well-presented and tested.

**Q9 Complying With Reviewing Instructions:**

1: Yes.

---

### Decision · Program_Chairs · 2022-05-15

**Decision:**

Accept (Oral)

**Comment:**

Meta Review: This is a really nice paper that has been praised by all the reviewers, and represents something of a breakthrough in the understanding of MAG equivalence classes.  It has a very high average score, and therefore I recommend it be given an oral presentation at the conference.

### Minor Comments
 - page 2: I think that $m \in O(n^2)$ is not right here.  I think you want $m \in \Omega(n^2)$, or just say $m \sim n^2$ if you don't want to introduce new notation.
 - page 3: I don't think 'signficant' effort has been put in to this problem (2 papers!)  Maybe 'discernible' would be more accurate here?
 - page 7: Again, I'm not sure 'crucial improvement' is the right phrase here.  Here 'significant' would be better used!